# Indoor Air Quality: Assessment of Dangerous Substances in Incense Products

**DOI:** 10.3390/ijerph18158086

**Published:** 2021-07-30

**Authors:** Gabriela Ventura Silva, Anabela O. Martins, Susana D. S. Martins

**Affiliations:** 1INEGI–LAETA, Institute of Science and Innovation in Mechanical and Industrial Engineering (INEGI)—Associated Laboratory for Energy and Aeronautics (LAETA), R. Dr. Roberto Frias, 400, 4200-465 Porto, Portugal; amartins@inegi.up.pt; 2INEGI, Institute of Science and Innovation in Mechanical and Industrial Engineering, R. Dr. Roberto Frias, 400, 4200-465 Porto, Portugal; smartins@inegi.up.pt

**Keywords:** indoor air quality, VOC, incense, carcinogenic, mutagenic and reprotoxic compounds

## Abstract

Indoor air pollution has obtained more attention in a moment where “stay at home” is a maximum repeated for the entire world. It is urgent to know the sources of pollutants indoors, to improve the indoor air quality. This study presents some results obtained for twelve incense products, used indoors, at home, and in temples, but also in spa centers or yoga gymnasiums, where the respiratory intensity is high, and the consequences on health could be more severe. The focus of this study was the gaseous emissions of different types of incense, performing a VOC screening and identifying some specific VOCs different from the usual ones, which are known or suspected to cause severe chronic health effects: carcinogenic, mutagenic, and reprotoxic. Thirteen compounds were selected: benzene, toluene, styrene, naphthalene, furfural, furan, isoprene, 2-butenal, phenol, 2-furyl methyl ketone, formaldehyde, acetaldehyde, and acrolein. The study also indicated that incense cone type shows a higher probability of being more pollutant than incense stick type, as from the 12 products tested, four were cone type, and three of them were in the group of the four higher polluters. Benzene and formaldehyde presented worrying levels in the major part of the products, above guideline values established by the WHO. Unfortunately, there are no limit values established for indoor air for all the compounds studied, but this fact should not exempt us from taking action to alert the population to the potential dangers of using those products. From this study, acetaldehyde, acrolein, furfural, and furan emerge as compounds with levels to deserve attention.

## 1. Introduction

It is now well established that indoor air pollution contributes significantly to the global burden of disease in the population [1]. In the presence of indoor sources, indoor contaminant concentrations are higher, sometimes 10 times higher (e.g., VOCs) than the respective outdoor air levels, regardless of the building location. Moreover, if until recently people spent about 90% of their time in confined spaces, distributed by the workplace, means of transport, home or leisure spaces, today, for a ratio of the population, that time reaches 99% or even 100% in the case of sick people at home or students and workers teleworking. Indoor Air Quality (IAQ) has thus become an even more relevant concern, since prolonged exposure to the same profile of potentially toxic substances, even at low concentrations, may affect human health, causing or aggravating diseases such as allergies, nose and skin irritation, asthma, and other airborne respiratory infections, chronic obstructive pulmonary disease, lung cancer, and cardiovascular disease.

Acting to improve IAQ in any space requires understanding the occupants and the dynamics of the spaces. One of the main parameters is, without doubt, the sources that can be so diverse as construction materials [2,3], furniture and decoration materials [4], consumer products [5,6], air conditioning systems [7], the occupants themselves and their activities [8,9,10]. In the last decades, regulatory efforts have improved, and many chemical compounds have been subjected to restrictions. Nevertheless, many other compounds appeared to replace the restricted ones.

This article presents some results obtained for incense, a product used indoors, at home, and temples, but also in spa centers or yoga gymnasiums, as some persons believe that incense emit favorable fragrances that can relieve stress and facilitate the attainment of physical, mental, and spiritual balance [11].

Previous studies show that incense burning emits many particles [12,13,14,15], and some studies present the chemical characterization of the particulate phase [11,16,17]. The results show high emissions levels, which demonstrate that the use of those products without adequate ventilation represents a risk for health.

In a study presented by Ho and Yu [18], a high concentration level of formaldehyde and acrolein was detected in indoor environments (temples and homes) where incense is burning, exceeding the World Health Organization (WHO) air quality guideline [1] of 100 μg/m^3^ for formaldehyde. Other aldehydes like acetaldehyde, benzaldehyde, furfural, glyoxal, and methylglyoxal were also identified.

Lee and Wang [13] studied 10 types of incense and found that benzene, toluene, methyl chloride, and methylene chloride concentrations increased significantly during the burning of the products. They also measured the concentration after burning and found that for some VOCs, the concentrations were even higher after burning than during burning, which implies that the human exposure period may be lengthy.

In 2008 a review article published by Lin et al. [19] revealed that when incense smoke pollutants are inhaled, they cause airway dysfunction and advise that incense smoke is a risk factor for elevated cord blood IgE levels. It has also been indicated to cause allergic contact dermatitis, and it has been associated with neoplasm.

In the study conducted by He et al. [20], they found that pregnant women who frequently smelled the incense burning at late pregnancy had an associated higher risk of hypertensive disorders and higher blood pressure levels. Wei et al. [21] found an association between household incense burning and delay in infant gross motor development.

Incense products are also used as mosquito repellents, being a way to repel insects during summer overnight in households across the world. In the study performed by Lu et al. [22], about 230 compounds divided by 14 classes of VOCs were found in the smoke of mosquito-repellent incense. The number and content of alkanes were the highest, followed by aromatic hydrocarbons and esters. In 2018, Wang et al. [23] had already carried out a study on mosquito repellent incenses and found that formaldehyde was the major component, accounting for 10 to 20% of the total amount of pollutants.

The focus of the present study was the gaseous emissions of different types of incense, performing a VOC screening, and identifying some other different than usual specific VOCs, which are known or suspected to cause severe chronic health effects: carcinogenic, mutagenic, and reprotoxic. Other compounds without classification but known to have fatal consequences were also scrutinized. From the VOC screening, several VOCs of different families were identified: aromatic hydrocarbons, alkanes, alkenes, ketones, aldehydes. Only a set of compounds causing concern, taking into consideration the ECHA (European Chemicals Agency) [24] classification, were selected for this study:benzene (carcinogenic and mutagenic);toluene (suspected to be reprotoxic);styrene (suspected to be reprotoxic);naphthalene (suspected to be carcinogenic);furfural (suspected to be carcinogenic);furan (carcinogenic, suspected to be mutagenic and SVHC);isoprene (carcinogenic and suspected to be mutagenic);formaldehyde (carcinogenic, suspected to be mutagenic and skin sensitizing);acetaldehyde (carcinogenic and suspected to be mutagenic);2-butenal (suspected to be mutagenic);phenol (suspected to be mutagenic);2-furyl methyl ketone (fatal if inhaled and fatal in contact with skin); andacrolein (fatal if swallowed or inhaled).

## 2. Materials and Methods

Twelve different incense products commercialized in Europe were studied: four products were cone type, and eight were stick type. The characteristics of the products are presented in Table 1. The values presented are the average of the measurements performed on three different samples. Mass values were obtained using a balance Kern and the dimensions using a caliper. The diameter of the cone incense is the diameter of the basis of the cone. The burning time is the burning time coincident with sampling time, adapted to the total burning time of the different products.

The test was performed in a test chamber according to ISO 16000-9 [25] and EN 16738 [26]. The test chamber, in stainless steel, had a volume of 1.0 m^3^ and was supplied continuously with clean air to complete two air changes per hour. The visual control of the burning behavior was performed using a webcam installed inside the chamber. The temperature, relative humidity, and oxygen levels were recorded continuously, using a Logger 175-H2 from Testo and a ToxiRAE Pro from RAE, respectively.

With the test chamber empty, VOCs and VVOCs were collected in tubes with Tenax TA and Carboxen 569. Formaldehyde, acetaldehyde, and acrolein were collected in cartridges impregnated with DNPH.

On the day of the test, five sticks/cones of the incense were chosen. Two sticks/cones were placed inside the test chamber, and three sticks/cones were kept outside the chamber in a control room as foreseen in the standard. The test started with lighting the sticks/cones (using a gas flame). After 5 min, (equilibration time), pollutants were collected from the chamber on average for 20 min (between 15 and 25 min depending on the product), in tubes with Tenax TA/Carboxen 569 and cartridges (from Waters) filled with silica gel coated with 2,4-dinitrophenylhydrazine (DNPH). Sampling was performed using the pump Apex Casella for VOCs and AirChek XR5000, SKC for low molecular aldehydes.

For VOC and VVOC identification and quantification, based on ISO 16000-6 [27], thermal desorption in line with gas chromatography coupled to a mass spectrometer detector (GC/MSD) was used. The GC used is from Agilent Technologies, model 7890A, and the mass spectrometer detector is also from Agilent, model 5975C. The thermal desorption system is from DANI, model TD Master. In the thermal process, the samples were desorbed at 300 °C for 10 min. Desorbed VVOCs and VOCs were first captured in a Tenax-TA-filled cold trap at −25 °C, and then quickly heated to 300 °C to introduce analytes to the GC. Compounds were separated in an HP-5MS capillary column (length: 50 m, inner diameter: 0.20 mm, film thickness: 0.33 μm) with helium (purity > 99.9995%) as the carrier gas. The quantification of the selected compounds was performed using the specific response factors. The standard solutions were prepared, weighing the pure compounds (analytical balance Scaltec) and diluting them with methanol. The correlation factor of the analytical calibration curve exceeded 0.99, and the limit of detection reached 0.47 μg/m^3^ for toluene and 0.30 μg/m^3^ for benzene. With an expanded uncertainty of 4% for toluene, the analytical method was linear in the range of 10 to 5000 ng. Total volatile organic compounds concentration (TVOC) was calculated for all compounds eluted between hexane and hexadecane, using the toluene response factor.

Formaldehyde, acetaldehyde, and acrolein were determined based on ISO 16000-3 [28]. After sample collection, the cartridges were sealed and refrigerated at 4 °C until analysis. Each cartridge was extracted with 5 mL of acetonitrile. The extracted solutions were analyzed by high-performance liquid chromatography (HPLC) using a chromatograph from Agilent Technologies, model 1220 Infinity LC. The column was a Reversed-Phase C18 (Zorbax ODS, 25 cm × 4.6 mm, 5 μm). The emission factor of the compounds was calculated based on the specific response factor of the analytical method. The standard solutions were prepared using the pure derivatized compounds with DNPH, through weight (analytical balance Scaltec), and dilution with acetonitrile. The correlation factor of the analytical calibration curve exceeded 0.99, and the average limit of detection reached 0.0046 μg/mL.

### 2.1. Reagents

The solvent used for VOCs standard solutions was methanol (Fisher Chemical, Loughborough, UK, 99.99%), and the highest quality possible of pure compounds were used: benzene (Sigma-Aldrich, MO, USA, >99.9%), toluene (Sigma-Aldrich, WI, USA, 99.5%), styrene (Sigma-Aldrich, MO, USA >99%), naphthalene (Sigma-Aldrich, MO, USA, 99.9%), furfural (Fluka, Switzerland, >99%), furan (Aldrich, MO, USA, ≥99%), isoprene (Aldrich, MO, USA, 99%), 2-butenal (Aldrich, MO, USA, ≥99.5%), phenol (Sigma-Aldrich, Switzerland, 99%), 2-furyl methyl ketone (Aldrich, MO, USA, 99%). The solvent used for aldehydes with low molecular weight solutions was acetonitrile (Sigma-Aldrich, MO, USA, 99.9%), and the derivatives compounds of 2,4-DNPH were: formaldehyde-2,4-DNPH (Supelco, PA, USA, 99.9%), acetaldehyde-2,4-DNPH (Supelco, PA, USA, 99.9%), and acrolein-2,4-DNPH (solution in acetonitrile, Aldrich, PA, USA, 99.9%).

### 2.2. Evaluation of the Results

The assessment of the test results from the emissions was performed according to EN 16739 [29]. The personal exposure levels were calculated through short-term peak concentration (*STPC*) and worst-case time-weighted average (*TWA*) and compared with relevant published indoor air limits for benzene, naphthalene, formaldehyde, toluene, and styrene. The *STPC* value (μg/m^3^) over the measurement period is given by the formula:(1)$$STPC=\frac{SER}{\left[RV*VR\right]}$$

The *TWA* value is calculated according to the formula:(2)$$TWA=\frac{SER}{\left[RV*VR\right]}*AUF$$
where:

*SER*—specific emission rate (μg/h)

*RV*—room volume (m^3^), assumed as 30 m^3^

*VR*—ventilation rate (h^−1^), assumed as 0.5 h^−1^

*AUF* (Average Use Factor)—(Exposure time h per day/24 (h)) × (Frequency of use (day)/7 (day)), being assumed the frequency of use 4 days per week and exposure time 1 h per day.

EN 16739 [29] assumes the frequency of use 4 days per week and exposure time 4 h per day in the case of candles. Considering incense is not used so much, it was considered only 1 h per day in the calculations of *TWA*.

The limit values established by the WHO for indoor air are presented in Table 2. To note that as benzene is a genotoxic carcinogen in humans, the WHO cannot recommend a safe level of exposure. Considering that the geometric mean of the range of the estimates of the excess lifetime risk of leukemia at a benzene air concentration of 1 μg/m^3^ is 6 × 10^−6^, the concentrations of airborne benzene associated with an excess lifetime risk of 1/1,000,000 is 0.17 μg/m^3^.

## 3. Results

Table 3 shows the average values of temperature, relative humidity, and oxygen in the test chamber before starting and during the test. Maximum values achieved for the parameters are presented, except in the case of oxygen, where the value presented is the minimum value reached.

Table 4 shows the average values of the emission factor of VOCs and VVOCs, selected in this study in the test chamber for the twelve products tested. Only compounds with concentrations above 2 μg/m^3^ are reported. Values below are stated as not detected (n.d.). Values of TVOC are also presented.

The personal exposure levels were calculated through short-term peak concentration (*STPC*) and worst-case time-weighted average (*TWA*) as defined previously, using Equations (1) and (2). Table 5 presents the *STPC* values, and Table 6 the *TWA* values obtained by calculation.

The analysis of the results obtained was performed by grouping the compounds according to their effects on human health: carcinogenicity, reprotoxicity, and mutagenicity. The focus will be on worst-case time-weighted average (*TWA*) values as these health effects are usually a consequence of long exposure. In the case of dangerous compounds with acute effects, the focus will be on the short-term peak concentration (*STPC*).

### 3.1. Carcinogenic Compounds

The compounds recognized as carcinogenic are benzene, furan, isoprene, formaldehyde, and acetaldehyde. Naphthalene and furfural are suspected to be carcinogenic. Figure 1 presents the *TWA* values of these compounds observed in the study.

It can be observed that the exposure levels are worrying for benzene, especially for Inc 1 and Inc 10. Only Inc 5 and Inc 7 show values of benzene below 1 μg/m^3^. It should be highlighted that all the values are above the guideline limit value derived, assuming an excess lifetime risk of 1/1,000,000 of 0.17 μg/m^3^ [1]. Formaldehyde also presents high values being the worst cases for Inc 1 and Inc 12 but closely followed by Inc 4, Inc 7, and Inc 8. However, the *TWA* values are below the guideline limit value of 100 μg/m^3^ [1]. Though, considering that the limit value established by the WHO is based on 30 min of exposure, the comparison should be performed with *STPC* values (see Table 5). In that case, only Inc 5 and Inc 11 are below the guideline value. Inc 1 also presents the higher values for furan and acetaldehyde, followed by Inc 9 and Inc 11. Isoprene was detected at a higher concentration in Inc 11, followed by Inc 10, Inc 1, Inc 12, and Inc 3. Concerning furfural, the higher emissions were detected again for Inc 1, followed by Inc 12 and Inc 7. Naphthalene was below the limit of detection in four incenses (Inc 1, Inc 2, Inc 5, and Inc 12), and values were relatively low in the other products, conducting to *TWA* values below 0.03 μg/m^3^, being therefore below the guideline value of 10 μg/m^3^. However, this is the value determined in the gaseous phase, and, probably, naphthalene would be detected in the particulate phase in higher concentrations. Overall, Inc 5 is the product with lower values of *TWA* for carcinogenic compounds.

### 3.2. Mutagenic Compounds

From the compounds detected, the only one recognized as mutagenic is benzene. Suspected to be mutagenic are furan, isoprene, formaldehyde, acetaldehyde, 2-butenal, and phenol. Figure 2 (on a different scale) presents the *TWA* values of these compounds observed in the study.

The data analysis, in this case, is similar to the previously done for carcinogenic compounds as some of them are the same. Benzene is again the most worrying compound, followed by formaldehyde. (E)-2-butenal was detected in seven products: Inc 2, Inc 5, Inc 6, Inc 9, Inc 10, Inc 11 and Inc 12. Phenol was also detected in seven products: Inc 1, Inc 3, Inc 5, Inc 6, Inc 8, Inc 9 and 12. The values are low concerning long-term exposure, but they contribute to the overall exposure.

### 3.3. Reprotoxic Compounds

The compounds recognized as reprotoxic are toluene and styrene. Figure 3 presents the *TWA* values of these compounds observed in the study.

Both compounds were detected in all products, being Inc 1 the incense with higher levels, followed by Inc 3 and Inc 11. Inc 5 and Inc 7 presented the lower concentration values. Both compounds present concentration levels below the guideline limit values established by the WHO.

### 3.4. Other Dangerous Compounds

In this group, acrolein (fatal if swallowed, fatal if inhaled) and 2-furyl methyl ketone (fatal if inhaled, fatal in contact with skin) were identified. Given the acute effect, the values under analysis are the short-term peak concentration (*STPC*), presented in Figure 4.

Acrolein was detected in all products, being Inc 1, the product with the higher value, and Inc 5, the product with the lower value. 2-furyl methyl ketone was detected in four products, although in low concentrations.

### 3.5. TVOC Emissions

TVOC emissions include all compounds eluted between hexane and hexadecane. Compounds like furan, isoprene, formaldehyde, acetaldehyde, and acrolein are excluded.

From the analysis of the results presented in Figure 5, it can be observed that Incense 1 and 12 have the highest emissions of TVOC, followed by incenses 10 and 11. Note that these products (1, 11, and 12) are cone type. The lowest values were observed for Inc 5 and Inc 7.

## 4. Discussion

### 4.1. Influence of the Type of Incense

It can be observed that incense 1 (cone type) presented high values for all the compounds, except for isoprene. After, the products with the worst performance were Incense 10, 11, and 12, although not for all compounds. The product with the best performance was Incense 5 (stick type). It should be noted that Incenses 1, 11, and 12 were cone type, which indicates that this type of incense is more pollutant than stick products. However, incense 10 (stick type) is the exception. From the present study, it can be concluded that incense cone type shows a great probability to be more pollutant than incense stick type.

### 4.2. Influence of the Size of Incense

The size of the incense, and its mass or volume, can also influence the level of pollution generated, as the time of burning is generally proportional to the size. In the family of stick products, it can be observed that Incense 5 presents the best performance and is the smaller one in terms of mass and volume. The largest stick product in terms of mass and volume is incense 8, but it is the second-worst product, being incense 10 the worst. Therefore, the size of the incense influences the emissions level, but this is not linear. Concerning cone type, it can be observed that Incense 1 presents the worst performance, but it is smaller in terms of mass and volume, which shows that in this case, the composition of the product is much more relevant to the nature of the emissions.

However, it should be taken into account that normally, the persons left the products burning until the end of their life, and in this study, the burning time was limited to a maximum of 25 min, for comparison purposes. Therefore, the bigger products, as they have a longer burning time, have more potential for pollution.

### 4.3. Safety on Use Incenses: Assessment of the Risk

Considering the compounds with guideline values, an assessment of the risk can be performed for the incense products and an analysis of the factors involved in the exposition.

Table 7 presents the values of *STPC* and *TWA* concentrations and the guideline values for those compounds.

Considering short-term peak concentration, it can be observed that the values for benzene are higher than the guideline value between 100 times (Inc 5) and 3600 times (Inc 1). This represents a higher lifetime risk for cancer. Values obtained for formaldehyde are also higher than the guideline value, except in the case of Inc 5 and Inc 11. The values are higher than guideline values between 1.2 times and 2.7 times (Inc 1). The other compounds were all below guideline values.

Considering the worst-case time-weighted average, it can be observed that the values for benzene are higher than the guideline value between 3 times (Inc 5) and 85 times (Inc 1). The other compounds were all below guideline values.

From these facts, it could be said that Inc 5 seems to be the least bad and Inc 1 the worst, as it represents a higher risk of exposure to carcinogenic compounds.

The short-term peak concentration was calculated using the scenario of a room with a volume of 30 m^3^ and a ventilation rate of 0.5 h^−1^. On the other hand, the worst-case time-weighted average was calculated assuming the frequency of use 4 days per week and exposure time 1 h per day. If the ventilation rate is increased, the concentration levels will decrease, as we can see in Figure 6 for formaldehyde. Figure 7 shows the predictable behavior of the concentration with the variation of ventilation rate for Inc 1 and Inc 2. Based on the mathematical functions that best fit those points, it is possible to calculate the ventilation rate necessary to decrease the levels of formaldehyde to acceptable levels. For example, for Inc 1 only with a ventilation rate of 1.37 h^−1^ it will achieve a concentration of 100 μg/m^3^. However, it should be stressed that people can also be exposed to formaldehyde from multiple sources. Many building products emit formaldehyde, increasing the consumer’s total exposure and overall risk. Therefore, using the precaution principle, the limit value for each product should be only a fraction of the guideline value for total exposure.

### 4.4. Recommendations for Users

The results obtained show that some of the products tested can represent a risk to health in terms of inhalation exposure. Then some considerations are made about what the consumer can do to minimize the exposure resulting from incense products and consumer products in general.

There are two main strategies for risk reduction, “source control”, where the nature or strength of the sources or even their existence indoors is removed, replaced, or moderated, and “exposure control”, essentially through ventilation. The first strategy is preferable as prevention is better than mitigation. This option is, however, in the hands of the manufacturers and policy makers who can force to decrease the contaminants present in incense products. Another way of controlling exposure is the restriction of the time spent in a particular contaminated space and, as the last solution, dilution with increased ventilation, which can be implemented by the consumer.

First of all, the consumer should act responsibly and comply with the instructions of the product labels. He must be aware that some factors of their personnel activity in using a product have consequences on the concentration to which he is exposed, but also his family, including children. The manufacturer should provide this information in the instructions for use.

The frequency of use of incense and the duration of use will have a consequence on pollutant concentration which has a direct impact on the exposure to the pollutant. The greater these factors are, the higher the risk. The consumer should reduce the burning time and use it as few times as possible.

The quantity of products used will have a direct consequence on the concentration of pollutants which has a direct impact on exposure to the pollutant. Increasing the amount would increase the risk.

It should be stressed that people can be exposed to the same chemical from multiple sources. Many of the substances are also found in a wide range of other products, increasing the consumer’s total exposures and overall risk.

Increasing the ventilation of a space is a fast way to dilute the concentration of a contaminant in the area where a product was used, assuming that the outdoor air is cleaner and will not increase the concentration of pollutants indoors. Increasing ventilation by opening the windows during and after burning incense will have a direct consequence on contaminant concentration, decreasing it, and would reduce the risk.

In the case of extreme conditions, as high temperatures, high levels of ozone, and high levels of particulate matter, special attention should be taken to the use of incense products. Those extreme conditions could potentiate higher levels of exposure, for example, to secondary pollutants resultant from chemical reactions that will not occur in normal situations.

The consumer should diminish the time spent in the space where the incense was burned to decrease the time exposure. If possible, children should be absent from those spaces.

Special attention should be paid to vulnerable people such as children and people with health problems (asthma, COPD, etc). The inhalation rates of children are of major importance. Because of their size, physiology, behavior, and activity level, inhalation rates of children differ from those of adults. Potential determinants of children’s susceptibility include the continuing process of lung growth and development, incomplete metabolic system, immature host defenses, high rates of infection with respiratory pathogens [31].

## 5. Conclusions

This study has shown that incense products are relevant sources of indoor pollution in terms of gaseous pollutants, and in particular of dangerous substances. The study also indicated that incense cone type shows a great probability to be more pollutant than incense stick type, as from the 12 products tested, three of the four higher polluters were of the cone type.

Benzene and formaldehyde present worrying levels in the major part of the products, above guideline values established by the WHO (2010) [1]. Unfortunately, no limit values are established for indoor air for all the compounds studied, but this fact should not exempt us from taking action to alert the population to the potential danger of using those products. From this study, acetaldehyde, acrolein, furfural, and furan emerge as compounds with levels to deserve attention.

Many dangerous compounds are controlled, as they are not reported often, and the establishment of guidelines is focused on compounds usually found in indoor air. This methodology can create a vicious circle, as the new studies tend to focus on compounds with guidelines established, and other compounds tend to be neglected. We expect to contribute with data on concerning compounds despite having been outside the concern of the legislators.

## Figures and Tables

**Figure 1 ijerph-18-08086-f001:**
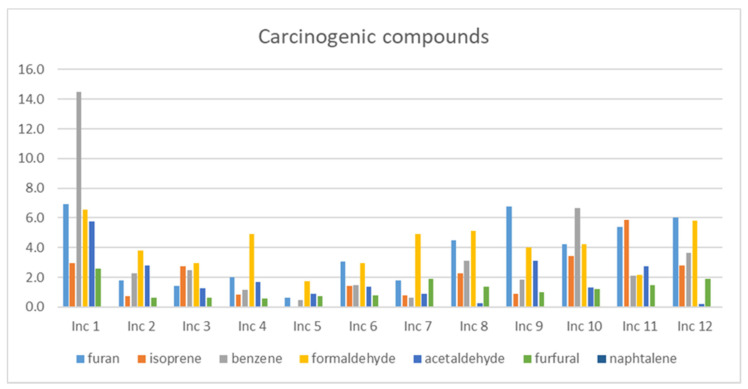
*TWA* levels (μg/m^3^) of the carcinogenic and suspected to be carcinogenic compounds in the twelve products studied.

**Figure 2 ijerph-18-08086-f002:**
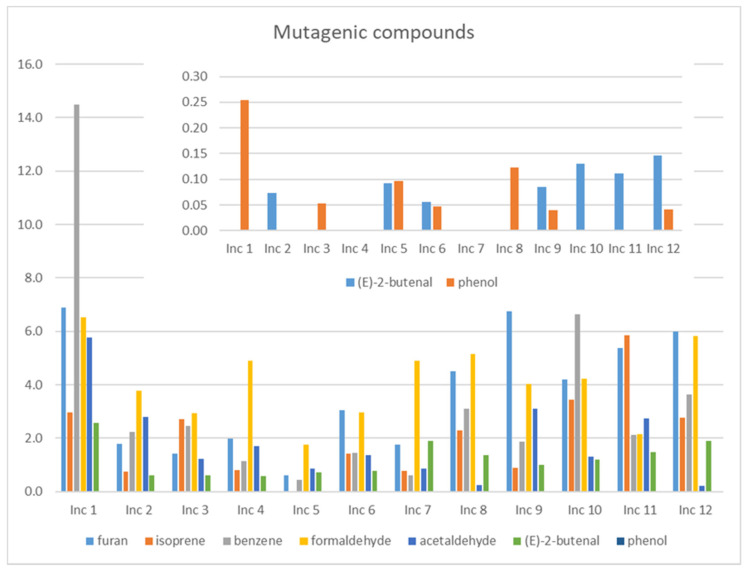
*TWA* levels (μg/m^3^) of the mutagenic and suspected to be mutagenic compounds in the twelve products studied.

**Figure 3 ijerph-18-08086-f003:**
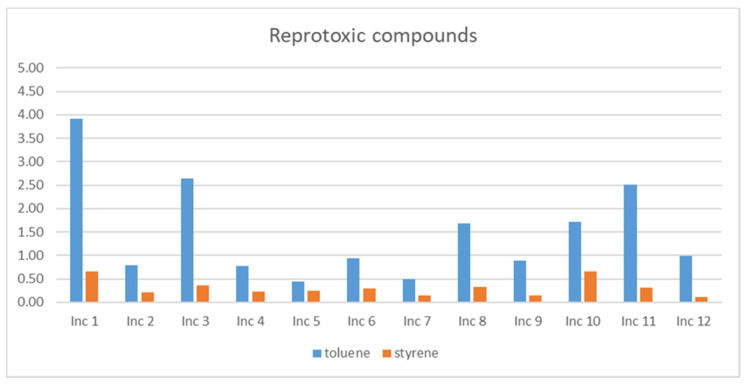
*TWA* levels (μg/m^3^) of the reprotoxic compounds in the twelve products studied.

**Figure 4 ijerph-18-08086-f004:**
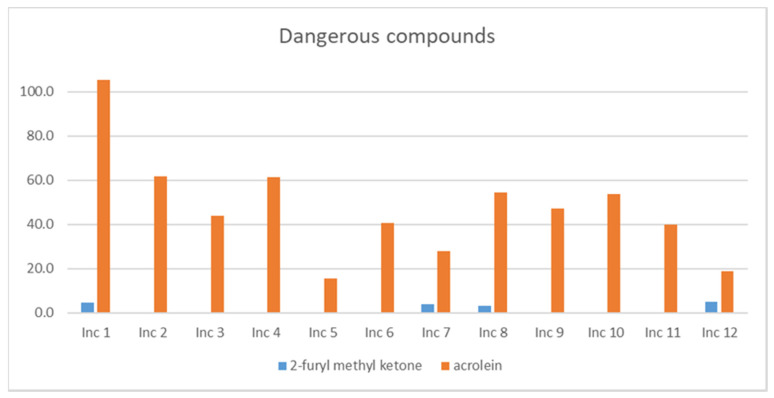
*STPC* concentrations (μg/m^3^) for other dangerous (fatal) compounds in the twelve products studied.

**Figure 5 ijerph-18-08086-f005:**
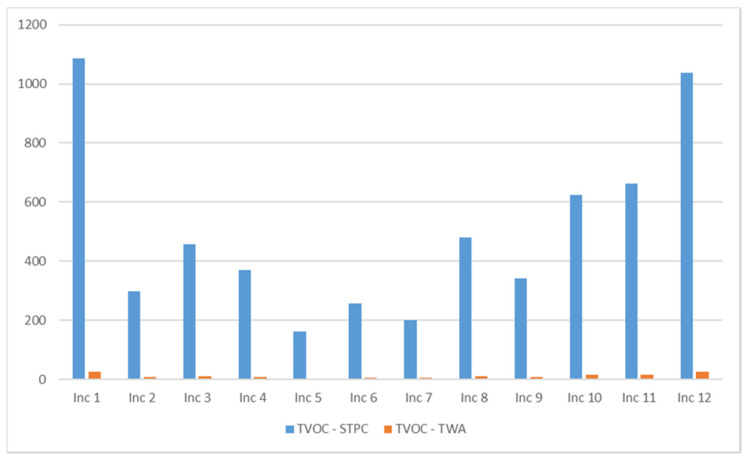
TVOC levels (μg/m^3^) in the twelve products studied.

**Figure 6 ijerph-18-08086-f006:**
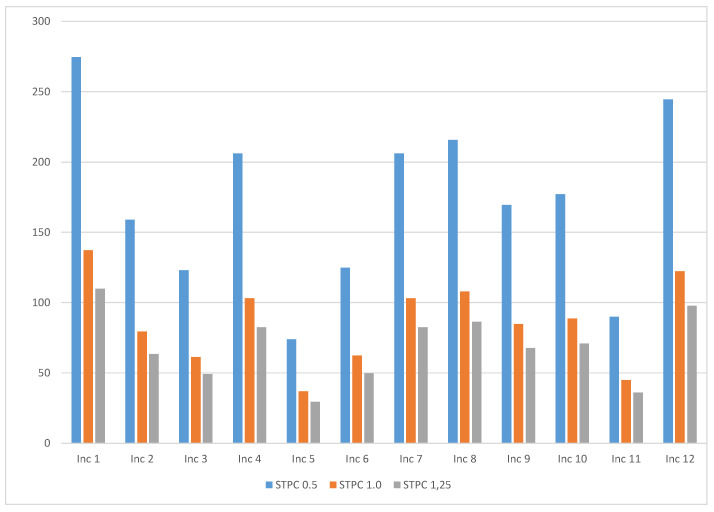
Average values of *STPC* (short-term peak concentration) (μg/m^3^) for formaldehyde assuming scenarios with different ventilation rates: 0.5 h^−1^, 1.0 h^−1^, and 1.25 h^−1^.

**Figure 7 ijerph-18-08086-f007:**
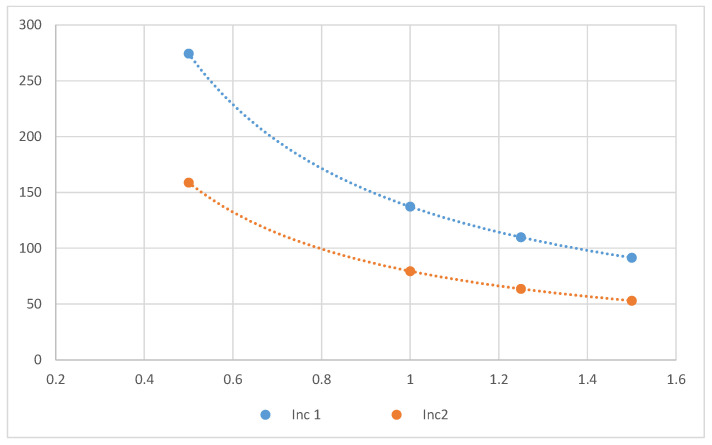
Better fit mathematical functions for values of *STPC* (short-term peak concentration) (μg/m^3^) for formaldehyde at different ventilation rates (h^−1^) for Inc 1 and Inc 2.

**Table 1 ijerph-18-08086-t001:** Physical characteristics of the incense products.

	Type	Length (cm)	Diameter (mm)	Volume (cm^3^)	Weight (g)	Burning Time (min)
		Combustible Part	Combustible Part			
Inc 1	Cone	3.42 ± 0.01	12.68 ± 0.12	1.44	1.68 ± 0.01	15
Inc 2	Stick	18.13 ± 0.21	3.12 ± 0.02	1.38	1.26 ± 0.06	20
Inc 3	Stick	14.87 ± 0.25	2.43 ± 0.19	0.69	0.77 ± 0.04	21
Inc 4	Stick	22.00 ± 0.24	2.67 ± 0.05	1.23	0.90 ± 0.01	20
Inc 5	Stick	13.37 ± 0.05	2.10 ± 0.00(0)	0.46	0.42 ± 0.00(5)	20
Inc 6	Stick	12.67 ± 0.12	3.25 ± 0.12	1.05	0.78 ± 0.04	20
Inc 7	Stick	15.63 ± 0.31	3.22 ± 0.18	1.27	1.23 ± 0.03	24
Inc 8	Stick	25.73 ± 0.12	3.97 ± 0.05	3.18	2.30 ± 0.18	25
Inc 9	Cone	3.19 ± 0.04	13.80 ± 0.11	1.59	1.30 ± 0.05	15
Inc 10	Stick	23.17 ± 0.05	3.73 ± 0.02	2.53	2.41 ± 0.06	25
Inc 11	Cone	3.48 ± 0.12	13.97 ± 0.05	1.78	1.71 ± 0.05	15
Inc 12	Cone	4.12 ± 0.04	18.03 ± 0.13	3.51	2.95 ± 0.02	25

**Table 2 ijerph-18-08086-t002:** Guideline values for individual substances [1,30].

Compound	Limit Value (μg/m^3^)	Averaging Time
benzene	0.17	associated with an excess lifetime risk of 1/1,000,000
formaldehyde	100	30 min
naphthalene	10	
toluene	260	1 week
styrene	260	1 week

**Table 3 ijerph-18-08086-t003:** Recorded values for Temperature, Relative Humidity and Oxygen.

	Inc 1	Inc 2	Inc 3	Inc 4	Inc 5	Inc 6	Inc 7	Inc 8	Inc 9	Inc 10	Inc 11	Inc 12	Average Values
Temperature (°C)													
Initial	24.2	24.0	24.5	24.0	24.0	22.9	23.4	23.3	22.9	22.9	24.0	23.5	23.6 ± 0.5
Final	25.0	24.4	25.1	24.4	24.5	23.3	24.2	23.5	23.5	23.8	24.5	24.6	24.2 ± 0.6
Maximum	25.6	24.5	25.1	24.5	24.7	23.4	24.2	23.6	24.2	24.0	25.0	25.3	24.5 ± 0.6
RH (%)													
Initial	49.8	50.2	47.2	45.8	46.4	48.9	48.1	49.7	49.8	47.9	41.8	45.8	47.6 ± 2.3
Final	50.2	50.2	47.1	46.6	45.9	49.1	48.2	49.5	49.7	47.2	42.0	45.2	47.6 ± 2.3
Maximum	52.0	50.7	47.3	46.8	46.5	49.4	48.3	49.7	50.3	47.4	42.5	45.6	48.0 ± 2.5
Oxygen (%)													
Initial	20.9	20.9	20.9	20.9	20.9	20.9	20.9	20.9	20.9	20.9	20.9	20.9	20.9 ± 0.0(0)
Final	20.9	20.9	20.9	20.9	20.9	20.9	20.9	20.9	20.9	20.9	20.9	20.9	20.9 ± 0.0(0)
Minimum	20.3	20.6	20.9	20.9	20.9	20.6	20.9	20.9	20.0	20.9	20.0	20.9	20.7 ± 0.3

**Table 4 ijerph-18-08086-t004:** Average emission factor (μg/h) values for TVOC, VOCs, and VVOCs selected in this study and observed in the test chamber for the twelve products tested.

	CAS	Inc 1	Inc 2	Inc 3	Inc 4	Inc 5	Inc 6	Inc 7	Inc 8	Inc 9	Inc 10	Inc 11	Inc 12
furan	110-00-9	4348	1133	898	1254	383	1924	1109	2838	4254	2653	3383	3780
isoprene	78-79-5	1868	470	1717	511	n.d.	889	497	1440	558	2172	3687	1752
(E)-2-butenal	123-73-9	n.d. *	45.9	n.d.	n.d.	58.1	34.9	n.d.	n.d.	53.9	82.5	69.9	92.1
benzene	71-43-2	9117	1412	1548	726	284	916	377	1964	1170	4180	1340	2300
toluene	108-88-3	2465	495	1663	486	281	589	312	1061	564	1081	1585	626
furfural	98-01-1	1616	386	392	363	463	497	1199	860	634	748	932	1189
styrene	100-42-5	413	136	228	145	151	187	95.7	204	89.7	417	195	66.2
2-furyl methyl ketone	1192-62-7	67.8	n.d.	n.d.	n.d.	n.d.	n.d.	56.7	46.7	n.d.	n.d.	n.d.	77.4
phenol	108-95-2	160	n.d.	33.0	n.d.	61.3	29.8	n.d.	77.3	25.4	n.d.	n.d.	26.1
naphthalene	91-20-3	n.d.	n.d.	6.01	3.00	n.d.	4.70	5.66	4.82	10.5	9.76	16.0	n.d.
formaldehyde	50-00-0	4117	2383	1845	3092	1108	1872	3091	3236	2540	2656	1350	3667
acetaldehyde	75-07-0	3634	1765	780	1067	547	864	544	155	1959	827	1731	139
acrolein	107-02-8	1581	927	658	920	235	608	421	817	707	807	601	284
TVOC		16310	4467	6866	5565	2433	3868	2986	7210	5115	9370	9928	15571

* n.d.—not detected means <2 μg/m^3^.

**Table 5 ijerph-18-08086-t005:** Average *STPC* values (short-term peak concentration) (μg/m^3^) for TVOC, VOCs, and VVOCs selected in this study in the test chamber for the twelve products tested.

	CAS	Inc 1	Inc 2	Inc 3	Inc 4	Inc 5	Inc 6	Inc 7	Inc 8	Inc 9	Inc 10	Inc 11	Inc 12
furan	110-00-9	290	75.6	59.9	83.6	25.5	128	73.9	189	284	177	226	252
isoprene	78-79-5	125	31.3	114	34.0	--	59.3	33.1	96.0	37.2	145	246	117
(E)-2-butenal	123-73-9	--	3.06	--	--	3.87	2.32	--	--	3.59	5.50	4.66	6.14
benzene	71-43-2	608	94.2	103	48.4	18.9	61.1	25.1	131	78.0	279	89.4	153
toluene	108-88-3	164	33.0	111	32.4	18.7	39.3	20.8	70.8	37.6	72.1	106	41.8
furfural	98-01-1	108	25.7	26.1	24.2	30.9	33.1	79.9	57.3	42.3	49.9	62.2	79.3
styrene	100-42-5	27.6	9.04	15.2	9.70	10.1	12.5	6.38	13.6	5.98	27.8	13.0	4.41
2-furyl methyl ketone	1192-62-7	4.52	--	--	--	--	--	3.78	3.11	--	--	--	5.16
phenol	108-95-2	10.7	--	2.20	--	4.09	1.98	--	5.15	1.69	--	--	1.74
naphthalene	91-20-3	--	--	--	--	--	--	--	--	0.70	0.65	1.07	--
formaldehyde	50-00-0	274	159	123	206	73.9	125	206	216	169	177	90.0	244
acetaldehyde	75-07-0	242	118	52.0	71.2	36.5	57.6	36.2	10.3	131	55.2	115	9.26
acrolein	107-02-8	105	61.8	43.9	61.4	15.6	40.6	28.1	54.5	47.1	53.8	40.1	18.9
TVOC		1087	298	458	371	162	258	199	481	341	625	662	1038

**Table 6 ijerph-18-08086-t006:** Average *TWA* values (worst-case time-weighted average) (μg/m^3^) for TVOC, VOCs, and VVOCs selected in this study in the test chamber for the twelve products tested.

	CAS	Inc 1	Inc 2	Inc 3	Inc 4	Inc 5	Inc 6	Inc 7	Inc 8	Inc 9	Inc 10	Inc 11	Inc 12
furan	110-00-9	6.9	1.8	1.4	2.0	0.6	3.1	1.8	4.5	6.8	4.2	5.4	6.0
isoprene	78-79-5	3.0	0.7	2.7	0.8	0.0	1.4	0.8	2.3	0.9	3.4	5.9	2.8
(E)-2-butenal	123-73-9	--	0.1	--	--	0.1	0.1	--	--	0.1	0.1	0.1	0.1
benzene	71-43-2	14.5	2.2	2.5	1.2	0.5	1.5	0.6	3.1	1.9	6.6	2.1	3.7
toluene	108-88-3	3.9	0.8	2.6	0.8	0.4	0.9	0.5	1.7	0.9	1.7	2.5	1.0
furfural	98-01-1	2.6	0.6	0.6	0.6	0.7	0.8	1.9	1.4	1.0	1.2	1.5	1.9
styrene	100-42-5	0.7	0.2	0.4	0.2	0.2	0.3	0.2	0.3	0.1	0.7	0.3	0.1
2-furyl methyl ketone	1192-62-7	0.1	--	--	--	--	--	0.1	0.1	--	--	--	0.1
phenol	108-95-2	0.3	--	0.1	--	0.1	0.0	--	0.1	--	--	--	0.0
naphthalene	91-20-3	--	--	--	--	--	--	--	--	0.0(2)	0.0(2)	0.0(3)	--
formaldehyde	50-00-0	6.5	3.8	2.9	4.9	1.8	3.0	4.9	5.1	4.0	4.2	2.1	5.8
acetaldehyde	75-07-0	5.8	2.8	1.2	1.7	0.9	1.4	0.9	0.2	3.1	1.3	2.7	0.2
acrolein	107-02-8	2.5	1.5	1.0	1.5	0.4	1.0	0.7	1.3	1.1	1.3	1.0	0.5
TVOC		25.9	7.1	10.9	8.8	3.9	6.1	4.7	11.4	8.1	14.9	15.8	24.7

**Table 7 ijerph-18-08086-t007:** Average values of *STPC* (short-term peak concentration) (μg/m^3^) and *TWA* (worst-case time-weighted average) (μg/m^3^) of VOCs with guideline values for the twelve products tested.

*STPC* (μg/m^3^)	Guideline Value (μg/m^3^)	Inc 1	Inc 2	Inc 3	Inc 4	Inc 5	Inc 6	Inc 7	Inc 8	Inc 9	Inc 10	Inc 11	Inc 12
benzene	0.17	608	94.2	103	48.4	18.9	61.1	25.1	131	78.0	279	89.4	153
toluene	260	164	33.0	111	32.4	18.7	39.3	20.8	70.8	37.6	72.1	106	41.8
styrene	260	27.6	9.04	15.2	9.70	10.1	12.5	6.38	13.6	5.98	27.8	13.0	4.41
naphthalene	10									0.70	0.65	1.07	
formaldehyde	100	274	159	123	206	73.9	125	206	216	169	177	90.0	244
*TWA* (μg/m^3^)	Guideline value (μg/m^3^)												
benzene	0.17	14.5	2.2	2.5	1.2	0.5	1.5	0.6	3.1	1.9	6.6	2.1	3.7
toluene	260	3.9	0.8	2.6	0.8	0.4	0.9	0.5	1.7	0.9	1.7	2.5	1.0
styrene	260	0.7	0.2	0.4	0.2	0.2	0.3	0.2	0.3	0.1	0.7	0.3	0.1
naphthalene	10												
formaldehyde	100	6.5	3.8	2.9	4.9	1.8	3.0	4.9	5.1	4.0	4.2	2.1	5.8

## Data Availability

Data is contained within the article.

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
