# Peer review of "Indoor Air Quality: Assessment of Dangerous Substances in Incense Products"

_ijerph, 2021, doi:10.3390/ijerph18158086_

Round 1

Reviewer 1 Report

The work uses standard procedures to detect VOC from burning incense.

Althoug, I think that incense is not of general use, in some paleces and temples its use is constant and so, knwoledge about the hazard it implies is important and so publication is justified.

Author Response

We would like to thank the reviewer for the availability to review the manuscript, for positively evaluate the study and for the comments.

Reviewer 2 Report

A report provided more and some important informations on incense burning and the indoor air quality.

English editing should be considered before publishing. 

Author Response

(The authors gave the same response as above.)

Reviewer 3 Report

Thank you for giving me the opportunity to review this manuscript. It is an interesting topic, and the results contribute to discussions about the importance of the indoor environment for Health.

Minor text editing for English clarification recommended; content otherwise good as it is.

Author Response

(The authors gave the same response as above.)

Reviewer 4 Report

This manuscripts tries to study the pollution levels of engaging pollutants (13 VOCs) present in european commercial indoor incenses for house, gyms etc being separate from outdoor atmosphere. In this approach, the idea is rather different and hence attractive for readers. Rare studies were devoted to INDOOR incense pollutions and this attribute is innovating. Yet about the title and somehow the methodolog, the terminology and performing of “Identifying” is too simple and not sufficient for the valuable JCR Journal of IJERPH. It is highly advised to use terminologies like: “Indoor Air Quality Assessment”, “Indoor Air Risk Assessment” or “Indoor Air Quality/Risk Management”. “Identifying” is the first step in environmental Quality/Risk management process. What about other steps like Metric Criteria, qualitative and quantitative assessment, ALARP notion, Risk reduction, Risk Control, Risk management, Risk Mitigation, Risk Acceptance and the most important CONTINUAL IMPROVEMENT. A comprehensive study should deal with or at least hint to these immense steps. Risk Assessment is necessary because the risk content of fatal materials like acrolein is not comparable at all to others -like mutagenic compounds like toluene. As well the risk content in fatals like between acrolein and 2furyl methyl Ketone are so diverse. So performing Risk Assessment is vital and undeniable.

Dear authors should highlight some criteria about 1) the existing incenses in use and 2) some suggestions for productions of safer incenses in future. As examples of existing incenses, the human health risk classifications should be handled for cigarette addicted, drug addicted, people with cardiovascular diseases, people with respiratory diseases like asthma and sportsman as well as keeping safe distance from the incense place. What should be the criteria for indoor room dimensions, ventilations and present people in the room? What should not be the people actions in rooms like heavy respiratory sports? What are the best time intervals or time frequencies for using present incenses, once a week, once a month etc? Indeed TWA and STEL have these notions in themselves as formulas –yet each case should be stated and calculated simultaneously for SER, VR and RV- however these values should be classified separately for preparing standards and reference values for usage. As an instance can one use incenses in a car –having a too small space? I mean What is the minimum amount for RV? What is the minimum amount for ventilation rate as a standard value? These criteria  as well as TWA itself should be classified by degrees: good, average, bad, …, and denoted by attractive tables and charts being stated as a reference standard.

 As examples of future incenses, dear authors can give suggestions like substituting materials, resizing of the incense sticks or cones, and noting the important health instructions on ads and the labels of the products.

Answering to some questions in text is necessary:

 1 What is the difference between proposed STPC and STEL (Short Term Exposure Limit) defined by ACGIH or TLV (Threshold Limit Value)?

2 Can dear authors expand their study to other indoor air pollution sources like cigarettes or paints or classroom whiteboard markers? What are their suggestions about other indoor air pollution sources? Because the most of present materials in this study are the same as other sources.

Altogether I guess the main approach to this manuscript is fair and the most important aspect is its inherent innovation. I prefer to accept the manuscript in the case ONLY IF dear authors prepare a standard or a protocol INDEX for Indoor Air Quality ASSESSMENT and MANAGEMENT as an Appendix format to receive higher impacts. It should include attractive and referable tables, figures and documentations.

Author Response

(The authors gave the same response as above.)
